# Influential Factors and Spatiotemporal Characteristics of Carbon Intensity on Industrial Sectors in China

**DOI:** 10.3390/ijerph18062914

**Published:** 2021-03-12

**Authors:** Ying Han, Baoling Jin, Xiaoyuan Qi, Huasen Zhou

**Affiliations:** School of Business Administration, Northeastern University, Shenyang 110169, China; yhan@mail.neu.edu.cn (Y.H.); 1710430@stu.neu.edu.cn (X.Q.); zhouhuasen@hbfu.edu.cn (H.Z.)

**Keywords:** carbon intensity, Spatial Durbin Model, spatiotemporal characteristic, spillover effect

## Abstract

Based on the extended STIRPAT model and panel data from 2005 to 2015 in 20 industrial sectors, this study investigates the influential factors of carbon intensity, including employee, industry added value, fixed-assets investment, coal consumption, and resource tax. Meanwhile, by expanding the spatial weight matrix and using the Spatial Durbin Model, we reveal the spatiotemporal characteristics of carbon intensity. The results indicate that Manufacturing of Oil Processing and Coking Processing (S7), Manufacturing of Non-metal Products (S10), Smelting and Rolling Process of Metal (S11), and Electricity, Gas, Water, Sewage Treatment, Waste and Remediation (S17) contribute most to carbon intensity in China. The carbon intensity of 20 industrial sectors presents a spatial agglomeration characteristic. Meanwhile, industry added value inhibits the carbon intensity; however, employee, coal consumption, and resource tax promote carbon intensity. Finally, coal consumption appears to have spillover effects, and the employee has an insignificant impact on the carbon intensity of industrial sectors.

## 1. Introduction

Global temperature rising is a phenomenon that urgently needs human attention; it affects the human public health system and causes the spread of certain infectious diseases. Meanwhile, the main reason for global climate change is carbon emissions [1]. In 1992, the United Nations Framework Convention on Climate Change committed to alleviating temperature rise and attempting to mitigate global warming [2]. In 1997, the Kyoto Protocol stipulated the greenhouse gas reduction targets of industrialized countries. In 2015, the Paris Agreement stipulated that the contracting parties will take further action to alleviate the rising trend of world temperature and ensure that the average temperature will increase by less than 2 °C compared with the average temperature before industrialization [3]. According to the United Nations Environment Programme (UNEP) data, in 2018, Chinese carbon emissions ranked first globally, accounting for more than a quarter of the world. It shows that China is facing tremendous pressure to reduce carbon emissions [4]. Industrial sectors are the backbone of the national economy, providing material guarantees for social production and life. At the same time, it will generate massive energy consumption, especially fossil energy. Notably, fossil energy combustion is indispensable in promoting carbon emissions. The data in Figure 1 reported that the carbon emissions produced by coal consumption in 1990 were 1791 million tonnes; however, in 2018, the carbon emissions grew to 7612 million tonnes, nearly 4.3 times compared with 1990. Moreover, with an average annual growth rate of 13.5%, the carbon emissions produced by crude oil consumption rose from 278 million tonnes to 1364 million tonnes from 1990 to 2018, which increased by four times. Meanwhile, carbon emissions of natural gas had changed slightly from 23 million tonnes in 2010 to 33 million tonnes in 2018. In addition, the carbon emissions produced by coal consumption occupied the largest share of the total in 2018, which was around 79% [5]. Furthermore, with the acceleration of industrialization, the carbon emissions emitted by industrial sectors have attracted more attention. In 2014, the National Climate Change Plan (2014–2020) first proposed the interim targets for reducing carbon emissions from the Steel and Cement industries by 2020. It was essential to control carbon emissions from high energy consumption industries, particularly Electricity, Chemical industries, Construction industries, and Steel Materials. Therefore, to gain insight into the potential for carbon reduction, it is meaningful to analyze the influential factors of carbon intensity in industrial sectors.

Industrial sectors provide the raw materials of industrialization infrastructure and are indispensable in the national economy [6]. Furthermore, economic development is dominated by industrial sectors [7]. The National Bureau of Statistics of China (NBSC) data showed that the total output value of Agriculture, Forestry, Fishery, and Animal Husbandry was 11,357.95 billion yuan, which accounted for 12.6% of the gross domestic product (GDP) in 2018. The industrial output was 3601.52 billion yuan, which accounted for 33.9% of the GDP. From 2001 to 2018, the average growth rate of Agriculture, Forestry, Fishery, and Animal Husbandry and Construction industry was 4.2% and 17.7%, respectively. From 2014 to 2017, the industry value added of the Power, Heating, and Gas Supply industry growth was from 1481.9 billion yuan to 1679.7 billion yuan [8].

Industrial sectors play significant roles in exchanging materials; apparently, the development of an industry will be affected by the environment [9]. Meanwhile, the relationship between industrial sectors is also undergoing profound changes. An industry cannot exist independently of the economy and other sectors, indicating the close connections between industries. Up to now, most of the studies have explored the spatiotemporal variations of carbon intensity from the level of national or provincial; however, few studies have paid close attention to the influential factors and spatial characteristics of carbon intensity from the aspect of industrial sectors. Additionally, many studies have decomposed the influential factors of the high-energy-intensive industries, such as the Power industry, Transportation industry, and the Coal industry [10]. However, the study on the spatial characteristics of carbon intensity at industrial sectors is current in the blank appearance.

Our study utilized the spatial panel econometric analysis model to analyze the influential factors and the spatial characteristics of carbon intensity on 20 industrial sectors in China. The contributions of our study include: (1) Compared with other literature, we analyze the influential factors of carbon intensity in the industrial sectors. (2) We expand the spatial distance matrix in the spatial analysis method to make it suitable for the spatial characteristics of industrial sectors and decompose the influential factors with the extended STIRPAT model. Given that few studies consider the impact on environmental regulations, this study takes advantage of the resource tax to express the role of environmental regulations. (3) We attempt to explore the spillover effect of carbon intensity with the spatial panel econometric analysis model from the perspective of industrial sectors.

The following contents are: Section 2 is the literature on carbon intensity. Section 3 presents the extended STIRPAT model, and spatial panel econometric analysis model, and the data source. Section 4 is the results and discussions. Section 5 draws to the conclusions and provides suggestions.

## 2. Literature Review

### 2.1. Influential Factors and Study Perspectives of Carbon Intensity 

At present, the influential factors of carbon emissions have been widely studied. Ref. [11] estimated Chinese industrial carbon emissions and decomposed the influencing factors into employees, energy structure, energy intensity, and economic effect. By decomposing carbon emissions into energy intensity, energy structure, and economic activities, ref. [12] indicated that the most positive influential factor of carbon emissions is economic activity. Ref. [13] studied the linkage within the urbanization, energy intensity, the proportion of industry value added in GDP, and energy structure. The results showed that technological innovation could raise the efficient utilization of energy and reduce energy intensity. Considering both macroeconomic and microeconomic perspectives, ref. [14] investigated the influencing factors of industrial carbon emissions, including the efficiency and intensity of R&D, the structure and intensity of energy, investment intensity, and economic activity. Notably, the existing research investigates the influential factors with energy structure, energy intensity, economic development, etc. Additionally, employees and industry value added are essential to assess the changes in carbon emissions. Meanwhile, only a few research studies considered the impact of environmental regulation on carbon intensity. 

Many scholars have studied carbon intensity from the national and regional levels, while some scholars have also analyzed the carbon intensity of industries [15,16,17]. For instance, ref. [18] identified the affecting mechanism of greenhouse gas emissions from national and regional levels in Korea. Using the extended STIRPAT model, ref. [19] explored the carbon emissions from the perspective of the province, the results found that population and industrialization promoted carbon emissions. Ref. [20] investigated the influencing factors of carbon intensity in Liaoning province, the results indicated that Construction Sectors were projected to contribute more carbon emissions. Ref. [21] analyzed the carbon emissions at the sectoral dimension of Jiangsu province, the study revealed that the most fundamental driving factor in carbon emissions was industrial output growth. For distinguishing the effects of macroeconomic policies on carbon intensity through changes in shares of industries, ref. [22] investigated carbon emissions from the Turkish economy in Agriculture, Industry, and Services. Furthermore, most researchers have studied the driving forces and spatial distribution of carbon emissions at multiple levels [23,24]. Taking national and provincial levels into consideration, ref. [25] calculated the direct carbon emissions and indirect carbon emissions produced by Power Generation and Heating Supply, respectively. Meanwhile, the results revealed the spatial distribution of carbon missions. Ref. [26] analyzed the influential forces of carbon intensity from the perspectives of sectoral and regional simultaneously.

Many scholars have dedicated significant efforts to studying the influential factors, evaluation method, and the characteristics of carbon intensity from multiple perspectives, which provided the basis for our study; however, few scholars have considered the interaction of carbon intensity among industrial sectors. 

### 2.2. Spatial Panel Data Model for the Study of Carbon Emissions

Panel data analysis is a method which involves regression from the dimensions of spatial and temporal. Ref. [27] revealed the spatial spillover effects of carbon intensity and highlighted the spatial and temporal evolution of carbon emissions at the city level. Basing on the provincial panel data and the extended STIRPAT model, ref. [28] studied the influential factors and spatiotemporal variations of carbon emissions. The study revealed that the driving factors were urbanization and economy; meanwhile, the industry proportion also promoted carbon emissions. However, the inhibiting factors were energy structure, energy intensity, and the development of technology. Ref. [29] explored the spatial changes of major factors contributing to carbon intensity across regions and concluded that partial-temporal analysis of carbon intensity was essential. Table 1 references the study period, perspectives, models, and influential factors of carbon intensity.

In general, the innovations of this study are in two aspects. First, we calculated carbon intensity at the level of 20 industrial sectors in China from 2005 to 2015; moreover, the extended STIRPAT method is utilized to study the influential factors of carbon intensity. Second, we employed the spatial panel econometric analysis model to examine the temporal and spatial variations of carbon intensity and additionally analyzed the direct effects and spillover effects of carbon intensity among industrial sectors.

## 3. Method and Date

### 3.1. Estimating Carbon Emissions in Industrial Sectors

Since there are no available data on carbon emissions in China, this study refers to the calculation method of the Intergovernmental Panel on Climate Change (IPCC) and estimated carbon emissions of 20 industries. The carbon emissions of industrial sectors are calculated by:(1)$$DCE_{i}^{t}=\sum_{j=1}^{8}E_{ij}^{t}*NCV_{j}*C_{j}*OF_{j}*f_{j}*\frac{44}{12}$$

Carbon intensity is the amount of carbon dioxide emitted per unit of GDP. Similarly, in industrial sectors, carbon intensity can be expressed with carbon emissions and industry added value. The expression of carbon intensity is as follows:(2)$$DCI_{i}^{t}=\frac{DCE_{i}^{t}}{IAV_{i}^{t}}$$
where *i* is the *i*th industrial sector; *j* denotes the *j*th energy fuel, which includes coal, coke, crude oil, gasoline, kerosene, diesel oil, fuel oil, and natural gas; *t* is the period from 2005 to 2015; $DCE_{ij}^{t}$ represents the carbon emissions of the *i*th sector, *j*th fuel and *t*th year; $E_{ij}^{t}$ stands for the primary energy consumption of *i*th sector, *j*th fuel and *t*th year; *NCV_j_* denotes the low calorific value of the *j*th fuel; *C_j_* denotes the coefficient of carbon content of the *j*th fuel; *OF_j_* and *f_j_* is the carbon oxidation and standard coal conversion coefficient of the *j*th fuel, respectively. $DCI_{i}^{t}$ represents carbon intensity on the *i*th sector and the *t*th year; $IAV_{i}^{t}$ is the industry added value of the *i*th sector and *t*th year. The conversion coefficient and CO_2_ emissions conversion coefficient of different kinds of energy are shown in Appendix A.

### 3.2. The Extended STIRPAT Model 

As outlined by [39], the IPAT model can be utilized to explore the influence of anthropogenic activities on the environment. The IPAT model is shown in Formula (3). I, P, A, and T represent the impact of environment, population size, affluence, and technology, respectively. Nevertheless, the IPAT model also has some limitations. For example, the IPAT model cannot deal with non-monotonic and non-propositional changes in variables. To overcome this shortcoming of the IPAT model, ref. [40] established the STIRPAT model as shown in Formula (4). The model is usually written in logarithmic expression, which is presented in Formula (5) below:(3)I=P*A*T
(4)$$I_{i}=aP_{i}^{b}A_{i}^{c}T_{i}^{d}e_{i}$$
(5)$$\ln I=\ln a+b(\ln P_{i})+c(\ln A_{i})+d(\ln T_{i}^{})+\ln e_{i}$$
where *a* is the model coefficient; *b*, *c*, and *d* are coefficients of variables, *e* denotes residual error; *i* denotes cross-sectional units.

Furthermore, the factors in the IPAT model can be disaggregated into further driving factors in environmental pressure; therefore, the model is extensively adopted to examine multiple influencing factors of carbon emissions [41,42]. Considering the characteristics of carbon emissions in the industrial sectors, we extended the STIRPAT model by the factors of the economic scale, energy structure, and environmental regulation. The extended STIRPAT model can be established as (6).
(6)$$\ln DCI_{it}=\alpha_{it}+\beta_{1}(\ln WP_{it})+\beta_{2}(\ln IAV_{it})+\beta_{3}(\ln FAI_{it})+\beta_{3}(\ln CR_{it})+\beta_{5}(\ln RTE_{it})+e_{it}$$
where *i* and *t* are the *i*th sector and *t*th year, respectively; $DCI_{it}$ denotes carbon intensity in the *i*th sector and *t*th year; $WP_{it}$, $IAV_{it}$, $FAI_{it}$, $CR_{it}$, and $RTE_{it}$ denote employee, industry added value, fixed asset investment, coal consumption, and energy resource tax, respectively; $\beta_{1},\;\beta_{2},\;\beta_{3},\;\beta_{4},\;\beta_{5}$ are the elasticity of each variable; $e_{it}$ is random error. 

Basing on the extended STIRPAT model, this study constructs a spatial panel data model to study the influencing factors of carbon intensity. Specifically, the explained variable is the carbon intensity of 20 industrial sectors. The explanatory variables are employee, industry added value, fixed-assets investment, coal consumption, and resource tax from 2005 to 2015. Meanwhile, all the economic indicators are converted into 2005 constant prices. In order to make the data more in line with the normal distribution and eliminate the heteroscedasticity of the model, this study performs a logarithmic transformation on the explained variable and the explanatory variable before regression [43,44].

The variables are defined as follows:

Employee (WP): is denoted by the number of working people in the industrial sectors. WP reflects the actual utilization of all labor resources within a certain period of time and is an indicator for analyzing the process scale of the industrial sector. To some extent, WP indirectly represents the scale and population intensity of an industry in the process of economic activities [45].

Industrial added value (IAV): is the economic output of industrial sectors. The level of economic development reflects the ability to create value, consumption, and investment in a country or region. Similarly, industry added value refers to the added value of unit output value in a certain period, which stands for the production capacity. Meanwhile, it is an important indicator that affects carbon intensity [46].

Fixed asset investment (FAI): is an essential comprehensive index for the state to prescribe an investment plan and control the investment scale. FAI refers to the number of assets invested by the unit in the current period when the service life of fixed assets exceeds one year. It includes the newly built civil engineering and the purchased equipment [47].

Coal consumption (CR): describes the coal consumption in the productive process of industrial sectors. Currently, the economic development of regional and industrial is inevitable to utilize fossil energy; therefore, it is indispensable to study the impact of fossil energy on carbon intensity [48].

Resource tax (RTE): represents environmental regulation. Environmental regulation is a vital embodiment of economic and environmental policies in a region or industrial sector. However, the effect of resource tax still needs further discussion. To observe the influence of environmental regulation on carbon intensity, we attempted to utilize resource tax as the supervision of environmental regulation in the process of industrialization [49].

### 3.3. Spatial Econometric Analysis Model

#### 3.3.1. Spatial Autocorrelation of Carbon Intensity

To further evaluate the spatial agglomeration effect on the carbon intensity of industrial sectors, the spatial correlation test with Moran’s I is essential. The global Moran’s I can be calculated as follows:(7)$$\begin{array}{l} \begin{array}{cc} Moran’s & I \end{array}=\frac{\sum_{i=1}^{n}\sum_{j=1}^{n}w_{ij}(X_{i}-\bar{X})(X_{j}-\bar{X})}{\frac{1}{n}\sum_{i=1}^{n}(X-\bar{X}{)}^{2}\sum_{i=1}^{n}\sum_{j=1}^{n}w_{ij}}\\ \begin{array}{cccc} & where & & \bar{X} \end{array}=\frac{1}{n}\sum_{i=1}^{n}X_{i} \end{array}$$
where $w_{ij}$ is the elements in the spatial weight matrix, $X_{i}$ denotes the observation variables in the *i*th industrial sector, and ${\overset{¯}{X}}_{i}$ is the average for carbon intensity. The distribution of Moran’s I index is −1 to 1. When Moran’s I > 0, it implies a positive spatial aggregation effect. Simultaneously, the closer the value to 1, the more significant the spatial correlation is. When Moran’s I < 0, it represents a negative aggregation effect. Meanwhile, the closer the value to −1, the more significant the spatial disparity is. Especially if Moran’s I = 0, the spatial distribution is random.

#### 3.3.2. Spatial Weight Matrix 

The input-output method proposed by [50] connected multiple industries into one system, reflecting the correlation between economics and industries. Therefore, the input-output method provides theoretical support for the extension of the spatial weight matrix between sectors. Ghosh established the Ghosh inverse matrix from the supply perspective, indicating that the industry needs to allocate its products to other industries while obtaining a certain amount of human, material, and financial resources [51]. The input-output relationship between industries is generated through both supply and demand. The complete consumption matrix obtained through the Leontief inverse matrix represents the correlation between industries driven by demand. The Ghosh inverse matrix and the identity matrix are used to get the complete distribution coefficient matrix to describe the correlation between sectors due to the supply relationship. Furthermore, the input-output table comprehensively reflects the input-output relationship among various national economies, and it also reveals the interdependent and mutually restricted relations among different industries. The input-output describes how each department obtains intermediate inputs and initial inputs from other departments for its production [52]. Meanwhile, it reveals the indirect and even neglected economic and technological links between various sectors, which is the quantitative basis of industrial structure research. However, the real economic trade relationship between the industry is quite complicated. This study innovatively uses the mean value of the complete consumption coefficient and the complete distribution coefficient to represent the relationship between sectors due to supply and demand, as is shown in Formula (8).
(8)F=0.5*[(L−I)+(G−I)]
where *L* stands for the Leontief inverse matrix; *G* stands for Ghosh inverse matrix; *I* stands for unit matrix; *F* is the correlation between industries generated by supply and demand.

The distance between industries can no longer be represented by regional geographic distance; therefore, the distance between sectors used in this study refers to the technical distance between sectors. While considering the economic distance and technological distance between industries, the space weight matrix of industries is shown in Formula (9).
(9)$$W_{ij}=\{\begin{array}{l} \begin{array}{cccc} \frac{f_{ij}}{{\left(k_{i}-k_{j}\right)}^{2}} & , & k_{i}\neq k_{j} & and\begin{array}{cc} & i\neq j \end{array} \end{array}\\ \begin{array}{ccc} & 1 & \begin{array}{ccc} & & k_{i}=k_{j}\begin{array}{ccc} & and & i\neq j \end{array} \end{array} \end{array}\\ \begin{array}{ccc} & 0 & \begin{array}{ccc} & & i=j \end{array} \end{array} \end{array}$$
where *i* and *j* represent the *i*th row and the *j*th column; $w_{ij}$ represents the elements of spatial weight matrix; $f_{ij}$ is the elements in the industry association relation matrix; $k_{i}-k_{j}$ stands for the technical distance of *i*th industrial sector and *j*th industrial sector; $k_{i}$ and $k_{j}$ are the level of technical utilization of energy, which can be calculated by the added value created by using 1 unit of energy consumption. 

#### 3.3.3. Spatial Panel Model

In the traditional panel model, the explanatory variables ignore the influences in other industries. For example, the carbon emissions of an industry are not only related to the carbon emission of “neighboring” industries, but may also be related to economic and social factors in other industrial sectors [53,54,55]. The spatial panel data model further considers the spatial lag explained variable WY and the spatial lag error term Wu. The former is called the spatial lag model (SLM), which mainly describes the spatial dependence (Equation (10)); while the latter is called the spatial error model (SEM), which reflects the spatial heterogeneity (Equation (11)). The main distinction between the SLM model and the SEM model lies in the different ways of describing spatial dependence [56]. However, when investigating the relationship between variables, the independent variable will also exit a spatial lag effect; moreover, this kind of spatial lag effect on the independent variable is often not negligible on the dependent variable [57]. In order to overcome limitations in the SEM and the SLM, the Spatial Durbin Model (SDM) incorporates WY and WX simultaneously (Equation (12)). The functions of SLM, SEM, and SDM were written as:(10)Y=ρWY+Xβ+μ+ϕ
(11)$$\begin{array}{l} Y=X\beta +\mu +\phi \\ \phi =\partial W\phi +\theta \end{array}$$
(12)Y=ρWY+Xβ+WXσ+μ+ϕ
where *Y* represents the matrix of the explained variables; *X* stands for the matrix of the explanatory variables; *W* is the spatial weight matrix of *N*N* (*N* is the number of industrial sectors); μ is the intercept item; β stands for the parameter vector of *X*; ρ and ∂ are the spatial regression coefficient of SLM model and SEM model, respectively. ϕ and θ represent the error term.

### 3.4. Data Source

(1)The categories and codes of the industrial sectors. Since there is a certain distinction between the national economic industries divided by the National Bureau of Statistics in China and the Organisation for Economic Co-operation and Development (OECD) input-output table, the two standards are considered and combined [58]. Twenty-two-digit industry names and codes in this study are as shown in Appendix A.(2)In the actual statistical process, the wide range of product exchanges between various industries and departments requires human resources, material resources, and time [59,60]. Therefore, in China, the corresponding input-output tables are only available in the years with mantissa 2 and 7, which directly results in the discontinuity of the input-output table. Due to the limitation of actual data, many studies using the input-output table to analyze practical problems can only be limited to some years. Considering that the input-output data before 2000 are too short of timeliness, this study only used the input-output table data after 2000. Meanwhile, the most recent year of the input-output table published by the China Input–Output Society is 2015, so the latest data used in this paper are from 2015 [61].(3)The data of WP, IAV, FAI, and CR were taken from the China Statistical Yearbook [62], China Industrial Statistics Yearbook [63], and the National Bureau of Statistics of China from 2005 to 2015 [8]. Furthermore, the data of RTE were obtained from the China Taxation Yearbook from 2005 to 2015 [64]. The statistical descriptions of variables are represented in Table 2.

## 4. Results and Discussions

### 4.1. Results of Carbon Intensity

To clearly describe the changes in the carbon intensity of 20 industrial sectors from 2005 to 2015, we divided the carbon intensity into three categories. The first category is the industrial sectors with a carbon intensity lower than 2 tonnes/10^4^ yuan, including S1, S3, S4, S5, S12, S13, S14, S15, S16, S18, and S19. In Figure 2, S1, S4, S5, S9, and S15 reach a peak in 2008 then show a downward trend; S13, S14, S16, S18, and S19 show fluctuating changes; however, the overall trend is declining; Meanwhile, the carbon intensity of these sectors reached its lowest points in 2011. Nevertheless, after 2011, the carbon intensity gradually increased and reached a second peak in 2013. The second category is the industrial sectors with a carbon intensity of 2–5 tonnes/10^4^ yuan. The carbon intensity of S6 presents a downward trend from 2005 to 2015. In the meantime, the carbon intensity of S2 shows a rebound trend, which drops to the lowest point in 2012. Moreover, the carbon intensity of S8 is fluctuant. Furthermore, the trend of S20 changes slightly; however, it presents an upward trend after 2014. The third category is the industrial sectors with a carbon intensity of more than 5 tonnes/10^4^ yuan. The carbon intensity of S7 is the highest among the 20 industries and shows an upward trend from 2005 to 2015. The total amount of carbon intensity remains at a high level, notwithstanding a slight downward trend in S10 and S11. At the same time, the trend of S17 is not apparent, which shows a fluctuating change. Overall, from 2005 to 2015, 20 industries present different trends in carbon intensity. S7, S10, S11, and S17 are the primary contributors to carbon intensity and should be the focus of carbon mitigation strategies.

### 4.2. Results of Spatial Autocorrelation Test

As illustrated in Table 3, from 2005 to 2015, the global Moran’s I is significant at the 1% level, which shows that the carbon intensity in China’s industrial sectors has spatial autocorrelation characteristics. Furthermore, the value of Moran’s I shows a fluctuating trend, indicating that the accumulation of carbon intensity in China’s industry is continually changing. Specifically, the aggregation degree of carbon intensity shows a decreasing trend from 2005 to 2007, then a rebound in 2008, while from 2009 to 2015, the trend of Moran’s I gradually decreased. The spatial autocorrelation test results further elucidated that it is significant to integrate the spatial effects into the analysis process to improve estimation accuracy.

The Moran index chart from right to left, from top to bottom, is HH quadrant, LH quadrant, LL quadrant, HL quadrant, where HH quadrant represents “High-High” agglomeration, and LL quadrant represents “Low-Low” agglomeration [65]. As shown in Figure 3, S7, S10, S11 and S17 mainly distribute in the HH quadrant, which accounts for 20% of all industries in 2005. S2 and S13 distribute in the LH quadrant, which accounts for only 10%; meanwhile, 70% of industries distribute in the LL. It indicates that the carbon intensity of 20 industrial sectors in 2005 has prominent spatial distribution characteristics. However, the results presented in Figure 4 reflect that, in 2015, the HH quadrant industries were S7 and S17, which decreased by two compared with 2005. The industries distributed in the LH quadrant were S2, S10, S11, and S13, which increased by two; Simultaneously, the number of industries in the LL quadrant remains unchanged compared to 2005. It finds that the accumulation of carbon intensity tends to weaken for the period 2005 to 2015.

### 4.3. Spatial Econometric Regression Results

#### 4.3.1. Test Results of the SDM Model 

Appendix A reflects that the independent variables are significant at the 1% level in the unit root test; it indicates no panel unit root in LnDCI, LnWP, LnIAV, LnFAI, LnCR, and LnRTE. The multicollinearity test of the explanatory variables is shown in Appendix A. The Variance Inflation Factor (VIF) values are all less than 10 verified that there is no multicollinearity on the explanatory variables. Furthermore, according to the Hausman test results (53.43, *p* = 0.000) in Appendix A, the panel data regression model is determined to be constructed using the spatial fixed form. To choose the appropriate spatial interaction effects of models, the LM test and the robust LM test are conducted on the non-spatial panel model [66].

The results in Table 4 denote that the coefficients of the LM spatial lag test on Pooled Ordinary Least Squares (OLS) and Time-period fixed effects are at the 1% significant level. The coefficients of Spatial fixed effects and Time-period fixed effects are at the 1% significant level of the LM spatial error test. Pooled OLS, Spatial fixed effects, and Time-period fixed effects are at the 5% significant level of the robust LM spatial lag; Further, the coefficients of robust LM spatial error are all at the 1% significant level. The LM test results and robust LM test confirm that the spatial panel models are more appropriate for the panel data analysis, and there exists a spatial correlation. Moreover, the results are following the conclusion of Moran’s I. To further illustrate which model is appropriate for the study, the Wald test and LR test were performed.

The results of Wald-lag (49.9291, *p* = 0.0003) and Wald-error (48.8836, *p* = 0.0005) show that the SEM model could not take the place of the SDM model; similarly, the results of LR-lag (44.9469, *p* = 0.0009) and LR-error (44.888, *p* = 0.0006) indicate that the SLM model could not replace the SDM model. Thereby, we apply the SDM model with Spatial fixed effects to interpret the results.

#### 4.3.2. Estimation Analysis of Spatial Durbin Model 

The results of the SDM model with spatial fixed effects are shown in Table 5. The coefficient of lnWP is significant at the 5% level, and the coefficients of lnIAV, lnCR, and lnRTE are significant at the 1% level. The industry added value is negative, and the other variables are all positive. The results illustrate that IAV has an inhibition effect on the industrial carbon intensity, while WP, CR, and RTE have a positive influence. Notably, ρ is significant at 5%, which demonstrates that the growth of one unit on the carbon intensity of an industrial sector can improve the carbon intensity in the neighboring industrial sector by 0.1720. It indicates that the carbon intensity of 20 industrial sectors has an apparent spatial spillover effect.

Specifically, the coefficient of WP is 0.1897; the result denotes that a unit growth in WP would lead to a 0.1897 increase in carbon intensity. The employed population can represent the labor production intensity of the industry. To some extent, the more the employed population indicates, the more production scale the sector has; therefore, the higher the consumption of fossil energy and the higher the carbon intensity.

The coefficient of CR is 0.0812, and it shows a positive effect. The result points out that a unit improvement in CR would cause a 0.0812 increase in carbon intensity. With the high-speed development of China’s industrial economy, the rationalization of structure on energy consumption is an inevitable choice to avoid the abuse of non-renewable energy and reduce carbon emissions. However, due to resource endowment constraints, the adjustment of coal-based consumption structure into clean energy is challenging in the short term. Therefore, it is indispensable to interiorize the concept of clean energy gradually.

The coefficient of RTE is 0.1177, showing that a unit promotion in RTE would lead to 0.1177 growth in carbon intensity. Generally speaking, resource tax has income effects and substitution effects on enterprises; the effects will affect the economic activities of enterprises. The purpose of resource tax is to achieve energy conservation and emission reduction. However, driven by economic interests, enterprises may prefer fossil energy at a competitive price, which increases carbon emissions and then enhances carbon intensity.

The coefficient of IAV is 1.0185, which reveals that a unit growth in IAV can bring a 1.0185 decrease in carbon intensity. Meanwhile, IAV has the largest negative effect on carbon intensity. From an economic point of view, carbon emissions are an externality. In economic activities, economic actors are concerned about the maximization of interests. Therefore, in the early stage of economic development, economic growth is usually achieved through large-scale consumption of resources so that carbon emissions will increase along with economic growth. With the continuous economic growth and technological progress, the economy pursues high-quality development, so it begins to control the extensive use of fossil energy; thus, the growth rate of carbon emissions slows down after the economic growth reaches a certain stage.

However, the results show that the FAI has no significance. Reasons may be that, on the one hand, China is a developing country with a larger proportion of investment in fixed assets based on construction, such as railway, highway, real estate, and other projects. In addition, some investments in industry and manufacturing projects have nontechnical and large pollution characteristics, which would drive the increase of coal consumption and enhance the difficulty of utilizing renewable energy. On the other hand, the increase in fixed asset investment would prompt the growth of the economy. Driven by economic expansion and the technology spillover effect, advanced production technology can be widely used, accelerating the reform of backward industries. Meanwhile, with the improvement of intelligence and industrialization, fixed-asset investment activities can form a production scale effect. In that way, the ecological and environmental quality will be improved. Therefore, the impact of fixed-asset investment on carbon intensity needs further discussion.

The results of W*lnWP, W*lnIAV, W*lnFAI, W*lnCR, and W*lnRTE are shown in Table 5. lnIAV and W*lnFAI are significant at the 1% level, and W*lnCR and W*lnRTE are significant at the 10% and 5% levels, respectively. The coefficient of W*lnFAI is negative, and the possible reason is that government macro-control plays a fundamental role in protecting and regulating the environment. By strengthening macro-control over fixed asset investment in neighboring industries, the government can effectively curb the over-rapid growth of investment and avoid excess production capacity, especially the overheated investment in large pollution industries.

The coefficient of W*lnRTE is also negative, and the possible illustrations are a series of policies issued by the government. For example, the energy tax and environmental tax are imposed on industries with high energy consumption and large emission characteristics. These policies are supposed to effectively control carbon emissions, thus reducing the carbon intensity of industrial sectors. The results illustrate the restraining effect of environmental regulation on the carbon intensity of China’s industrial sectors.

Significantly, W*lnIAV presents a positive effect, implying that the growth of IAV in a certain industrial sector would increase the carbon intensity of neighboring industries. With the growth of industry value added, the industries will be better able to exploit clean energy or produce materials, which lead to higher prices for clean energy. Therefore, the neighboring industries are more willing to use cheaper fossil fuels, which gives rise to carbon intensity growth. Consequently, it is essential to control the price of clean energy and maintain the stability of the market.

Similarly, the positive effect between carbon intensity and W*lnCR demonstrates that a higher CR in the neighboring industrial sector could strengthen the carbon intensity of the certain industrial sector. As neighboring industries increase the consumption of fossil fuels, the carbon intensity may exceed the warning line. Hence, the carbon intensity of adjacent industries is transferred to other similar industries, such as trading carbon emissions allowances, which leads to the growth of carbon intensity in similar industries.

#### 4.3.3. Results of the Direct and Spillover Effects

Basing on the SDM model, this study further calculates the direct and indirect effects of WP, IAV, FAI, CR, and RTE on the carbon intensity of industrial sectors. Three effects of the SDM model with spatial fixed effects are presented in Table 6. Based on the results of estimated coefficients, the variables can be classified into two categories.

The first category refers to lnIAV, lnCR, and lnRTE, simultaneously having direct and indirect effects. The direct and indirect effects of lnCR exert a significantly positive influence, indicating that coal consumption holds a promoting impact on carbon intensity. Meanwhile, the indirect effect is larger than the direct effect, which shows that the spatial spillover effect is more significant. Specifically, the coefficient of direct effect and indirect effect of lnCR are 0.0859 and 0.0909, with a 1% and 5% significance level. The results emphasize that coal consumption is the critical reason that gives rise to the increase in carbon intensity, and it is the priority factor to achieve carbon intensity reduction. However, there are differences in the direct and indirect effects of lnIAV. The direct effect of lnIAV is significantly negative; nevertheless, the indirect effect is significantly positive. The possible explanations are that with the increase of industry added value, the industry has significantly reduced the carbon intensity by investing in advanced technology and environmentally friendly equipment. However, there is competition among industries for patents, advanced technology, and equipment. The neighboring industry could not share the advanced equipment and technology; therefore, the carbon intensity would increase relatively. Besides, it is interesting that the direct and indirect effects of lnRTE are evident differences. The direct effect of lnRTE is 0.11307; however, the spillover effect is −0.1047. The interpretation is that the contribution of environmental regulation has not yet been adequately realized. Notwithstanding that the resource tax has been raised, the energy structure of industries has not been transformed, and fossil energy consumption is still dominant. Additionally, the benefits of using fossil fuels are more attractive to the industry. As China’s environmental regulation is not mature enough, resource tax standards between sectors are not perfect. For instance, the standards are inconsistent with the high carbon intensity of S7, S10, S11, and S17, which will lead to distinguishing effects of the resource tax.

The second category is lnFAI, which only has an indirect effect; meanwhile, the spillover effect of lnFAI is −0.2947. The fixed assets investment of the traditional industries with high energy consumption and heavy pollution will increase carbon intensity. However, the industries with a small fixed-asset investment are restricted by other similar industries. Under the effect of the market mechanism, the production scale of these industries is challenging to expand; thereby, the carbon intensity is relatively smaller. The results show that there are differences in the affecting mechanism of factors on industrial sectors. Notably, more targeted strategies are essential to restrict the carbon intensity of the industrial sectors in China.

## 5. Conclusions

Based on panel data, we estimated the carbon intensity and investigated the salient influencing factors of 20 industrial sectors from 2005 to 2015 in China. Meanwhile, the SDM model is verified to study the spatial-temporal differences in carbon intensity and reveal the spatial spillover effects. The major conclusions are as follows: First, from 2005 to 2015, S7, S11, S10, and S17 are the dominating contributors to carbon intensity, for the reason that those industries have a strong demand for coal consumption and the unreasonable energy structure, which belong to the traditional heavy industries. Second, the global Moran’s I index of 20 industrial sectors from 2005 to 2015 is positive. It indicates that China’s carbon intensity has the feature of spatial aggregation; moreover, HH and LL clusters are the principal types of aggregation. Meanwhile, S7 and S17 are always in the HH quadrant, which proves that there would be the highest potential reduction of carbon intensity. Third, the influence factors of direct and spillover effects on the carbon intensity are complicated. On the one hand, industry added value exits a restraining impact on carbon intensity. Simultaneously, the employee, coal consumption, and resource tax have different degrees of positive effect on the carbon intensity. On the other hand, industry added value, coal consumption, and resource tax have direct and indirect effects. In particular, coal consumption has noticeable spillover effects.

This study obtained the following policy suggestions through empirical research. Firstly, China is undergoing great pressure to implement strategies for controlling carbon intensity. The results show that the carbon intensity of S7, S11, S10, and S17 remains the trend of increase from 2005 to 2015. Therefore, we should pay widespread concern to the emission standard of carbon intensity in energy-intensive industries. Secondly, it is necessary to attach the vital role of strengthening cooperation between sectors. According to Moran’s I and the SDM model results, there is a spatial effect on industry carbon intensity from 2005 to 2015. The government should adopt the strategy of resource sharing, industrial cooperation, and information sharing. Thirdly, the SDM model results show that coal consumption has an evident spillover effect. Due to the consistency of technological process and coal utilization in similar industries, fossil energy consumption in one sector would cause a simultaneous increase in fossil energy consumption in other sectors. Therefore, the coordination of the related sectors should be fully considered to drive clean energy usage in their upstream and downstream industries. Finally, the direct and indirect effects of resource tax are distinguished. Generally speaking, strengthening government supervision and environmental regulation would undoubtedly reduce the carbon intensity of industrial sectors. However, the Chinese carbon tax policy is not mature enough to fully consider the characteristics of various industries. Therefore, implying differentiated resource tax according to the energy demand and energy efficiency of enterprises is essential.

This study makes exploratory use of spatial economic distance and technical distance between industries to establish the space weight matrix of sectors. However, decided by the input-output tables in China, the timespan of this paper is 2005–2015. In future studies, the ending year can extend. Moreover, the spillover effect of resource tax (RTE) in Chinese industrial sectors needs a detailed investigation.

## Figures and Tables

**Figure 1 ijerph-18-02914-f001:**
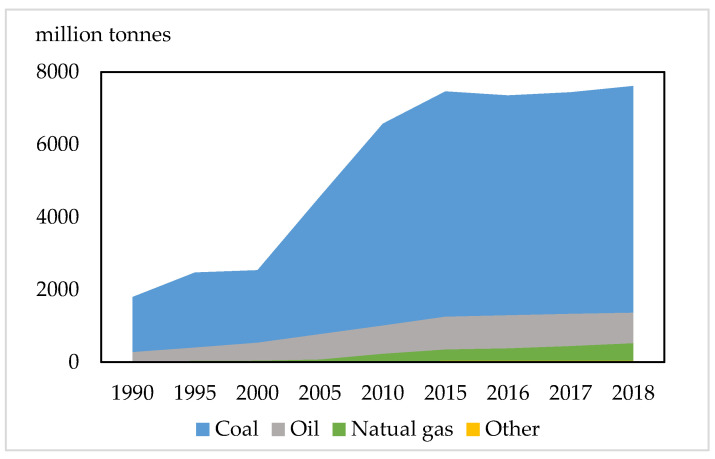
CO_2_ emissions by energy source from the International Energy Agency (IEA), China 1990–2018.

**Figure 2 ijerph-18-02914-f002:**
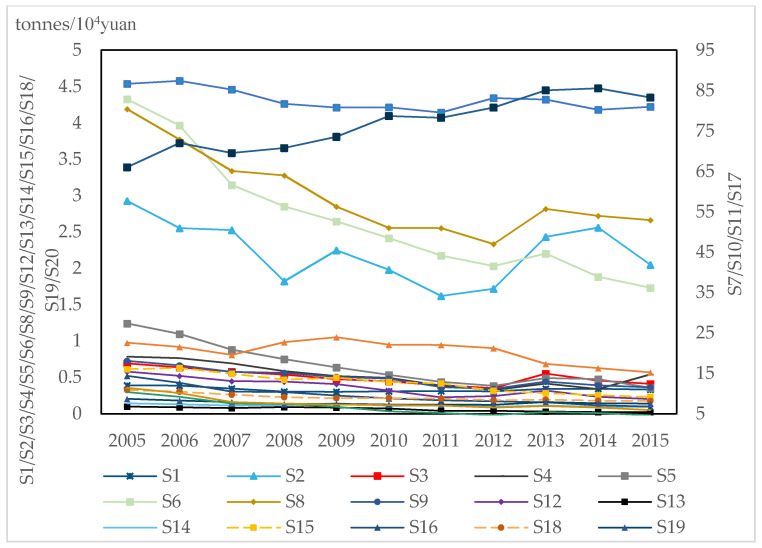
The carbon intensity of 20 industries.

**Figure 3 ijerph-18-02914-f003:**
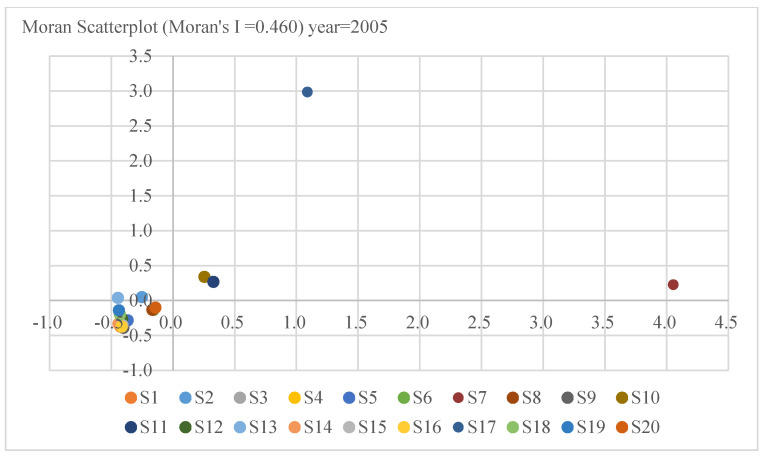
Moran scatters of carbon intensity in China in 2005.

**Figure 4 ijerph-18-02914-f004:**
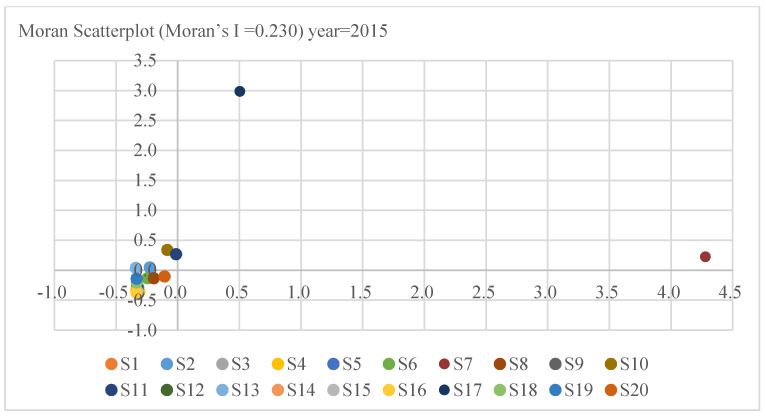
Moran scatters of carbon intensity in China in 2015.

**Table 1 ijerph-18-02914-t001:** Literature for influential factors of industrial sectors.

Reference	Period	Perspectives	Models	Influential Factors
[30]	1990–2006	Agriculture, Manufacturing, Service	Environmental Data Envelopment Analysis (DEA), Multiplicative LMDI	Energy intensity, Energy efficiency, Economic effect, Structural effect, Technical change, etc.
[31]	2000–2009	Agriculture, Industry, Construction, Transportation, Commercial, The Other Sectors	LMDI of sectoral	Energy intensity, Energy Structure, Structure effect, GDP, etc.
[32]	1985–2014	Transportation	LMDI	Energy intensity, Structure effect, Economic output, Population Effects
[33]	1985–2011	Power Industry	Scenario analysis, Monte Carlo analysis, LMDI	Population, Economic activity Energy efficiency, Electricity generation structure, Electricity intensity
[34]	2000–2014	Equipment Manufacturing Industry	Tapio decoupling, Evaluation model, LMDI	The average number of labor, Energy Intensity, Energy consumption, Industry value added
[35]	2002–2012	42 industrial sectors	SDA and IDA	Intermediate input, Added value, Total input, Energy structure
[36]	2016–2050	Building Sector	Emission reduction potential model, scenario analysis	Population, Urbanization rate, Total area, Rural building area, Commercial building area, Energy intensity, Energy consumption, Urban building area,
[37]	2011–2050	Iron and Steel, Electric, Power, Cement, Transport, Construction, Other industries	The ZSG-DEA model Scenarios analysis	GDP growth rate, Total GDP, Energy consumption growth rate, Total energy consumption,
[38]	2005–2015	26 industrial sectors	Input-output model	Propensity to consume, Population, Per capita income, Production intensity, Capital investment, Export

**Table 2 ijerph-18-02914-t002:** The statistical descriptions of variables.

Variables	Unit	Mean	Max	Min	Std. Dev	Median	Skewness	Kurtosis
Ln(DCI)	tonnes/10^4^ yuan	−0.0132	4.4495	−3.7907	1.8452	−0.6179	0.5880	−0.2711
Ln(WP)	10^4^ people	6.0868	7.9800	4.3095	0.7448	6.2405	−0.4296	0.0351
Ln(IAV)	10^9^ yuan	9.0985	10.9315	6.3130	0.9252	9.2664	−0.5674	−0.2150
Ln(FAI)	10^9^ yuan	8.4921	10.7991	5.6380	1.0331	8.5361	−0.1890	−0.6207
Ln(CR)	10^4^ tonnes of standard coal	7.6576	11.8227	4.6635	1.9001	7.2513	0.4224	−0.9689
Ln(RTE)	10^4^ yuan	10.2803	14.4714	6.7124	1.8813	9.9873	0.2508	−0.9226

**Table 3 ijerph-18-02914-t003:** The results of global Moran’s I.

Year	I	E(I)	Sd(I)	Z	*p*-Value
2005	0.460	−0.053	0.096	5.337	0.000
2006	0.400	−0.053	0.085	5.321	0.000
2007	0.372	−0.053	0.079	5.380	0.000
2008	0.425	−0.053	0.086	5.567	0.000
2009	0.430	−0.053	0.086	5.640	0.000
2010	0.371	−0.053	0.076	5.566	0.000
2011	0.372	−0.053	0.076	5.576	0.000
2012	0.347	−0.053	0.072	5.553	0.000
2013	0.262	−0.053	0.060	5.225	0.000
2014	0.242	−0.053	0.057	5.156	0.000
2015	0.230	−0.053	0.055	5.130	0.000

**Table 4 ijerph-18-02914-t004:** Results of the non-spatial panel model.

Variables	Pooled OLS	Spatial Fixed Effects	Time-Period Fixed Effects	Spatial and Time-Period Fixed Effects
lnWP	0.007117	0.110081	−0.062697	0.077985
	(0.070536)	(1.230155)	(−0.696842)	(0.907808)
lnIAV	−0.673708 ***	−0.930919 ***	−0.738112 ***	−1.075756 ***
	(−7.232953)	(−8.98873)	(−8.918925)	(−9.731367)
lnFAI	−0.526065 ***	−0.004036	−0.193378 *	−0.090964 *
	(−5.067212)	(−0.09171)	(−1.797468)	(−1.78732)
lnCR	0.298374 ***	0.071815 ***	0.24465 ***	0.060327 ***
	(5.423242)	(3.142763)	(4.960244)	(2.739103)
lnRTE	0.731265 ***	0.087624 ***	0.776671 ***	0.082699 **
	(10.047195)	(3.219515)	(11.845452)	(2.319146)
intercept	0.738175			
	(1.21517)			
R^2^	0.8119	0.7328	0.8509	0.4288
adj.R-sq	0.8075	0.7278	0.8482	0.4182
σ^2^	0.6584	0.0204	0.5127	0.0183
Durbin–Watson	1.8483	1.5975	2.2642	1.8401
Log-likelihood	−263.1514	118.5949	−236.1576	130.6136
LM spatial lag	63.0448 ***	1.2834	37.8617 ***	0.2227
LM spatial error	0.6931	7.8131 ***	113.4589 ***	0.0156
Robust LM spatial lag	112.6474 ***	6.1047 **	8.1296 ***	0.3682
Robust LM spatial error	50.2956 ***	12.6344 ***	83.7268 ***	0.1610

Notes: The symbols *, **, and ***, represent the significance at 10%, 5% and 1%, respectively.

**Table 5 ijerph-18-02914-t005:** Estimation results of the SDM model.

	Time Period Fixed Effects	*t*-Stat	Spatial Fixed Effects	*t*-Stat	Spatial and Time Period Fixed Effects	*t*-Stat	Spatial Random Effects and Time Period Fixed Effects	*t*-Stat
lnWP	0.16781 ***	(2.792187)	0.189684 **	(2.146972)	0.180104 **	(2.177192)	0.153244 *	(1.774003)
lnIAV	−0.297274 ***	(−5.233159)	−1.018526 ***	(−8.908154)	−1.083282 ***	(−10.191193)	−1.035031 ***	(−9.434184)
lnFAI	0.107959	(1.33415)	0.01054	(0.183685)	−0.007189	(−0.133335)	0.006709	(0.119745)
lnCR	0.100926 ***	(2.914759)	0.081244 ***	(3.570684)	0.058751 ***	(2.658103)	0.074476 ***	(3.228029)
lnRTE	0.22668 ***	(4.255919)	0.117734 ***	(3.479601)	0.082713 **	(2.383868)	0.120634 ***	(3.387609)
W*lnWP	−0.545619 ***	(−3.262531)	−0.002428	(−0.011857)	−0.036972	(−0.176001)	−0.177213	(−0.815479)
W*lnIAV	−0.352355 ***	(−2.788412)	0.803933 ***	(4.56538)	0.232655	(1.052079)	0.398847 *	(1.796506)
W*lnFAI	0.475544 ***	(3.075947)	−0.260073 ***	(−3.336868)	−0.317381 ***	(−4.166166)	−0.297505 ***	(−3.706654)
W*lnCR	0.49059 ***	(7.783382)	0.065306 *	(1.820832)	0.010556	(0.288807)	0.032331	(0.843988)
W*lnRTE	0.111858	(1.336613)	−0.111432 **	(−2.302324)	−0.235546 ***	(−2.907783)	−0.127943	(−1.605383)
ρ	0.082039	(1.218762)	0.172016**	(2.340959)	0.016039	(0.206813)	0.137008 *	
R^2^	0.9478		0.995		0.9954		0.9948	
Corr-squared	0.9461		0.7659		0.5012		0.1577	
σ^2^	0.1871		0.0189		0.0156		0.0175	
Log-likelihood	−122.72521		133.31188		145.37759		59.152713	

Notes: The symbols *, **, and ***, represent the significance at 10%, 5% and 1%, respectively.

**Table 6 ijerph-18-02914-t006:** Three effects of the SDM model with spatial fixed effects.

	Direct		Indirect		Total	
	Coefficient	*t*-Stat	Coefficient	*t*-Stat	Coefficient	*t*-Stat
lnWP	0.191349 **	(2.090209)	0.043921	(0.18183)	0.23527	(0.853011)
lnIAV	−0.983596 ***	(−8.958051)	0.717703 ***	(3.760386)	−0.265893	(−1.247669)
lnFAI	−0.005691	(−0.09889)	−0.29469 ***	(−3.220505)	−0.300381 **	(−2.744674)
lnCR	0.085909 ***	(3.576337)	0.090861 **	(2.146368)	0.17677 ***	(3.121794)
lnRTE	0.113068 ***	(3.498257)	−0.10465 *	(−2.021752)	0.008418	(0.169586)

Notes: The symbols ***, **, and * denote the significance at 1%, 5%, and 10%, respectively.

## Data Availability

All the tables and figures are made by the authors. The data are taken from the China Statistical Yearbook, China Industrial Statistics Yearbook, National Bureau of Statistics of China, China Taxation Yearbook. The data in this paper can be obtained from the authors.

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
