# Peer review of "Influential Factors and Spatiotemporal Characteristics of Carbon Intensity on Industrial Sectors in China"

_ijerph, 2021, doi:10.3390/ijerph18062914_

Round 1
Reviewer 1 Report
"affecting factors" does not sound grammatically correct, i would refer to as "Contributing factors" or "Influential factors"
line 10: ...this paper study investigated...
line 22: "Temperature rising" is too vague, state if you are referring to the environmental or global temperature increases.
Line 23: you state the increase is due to carbon emissions, presented as a fact therefore it requires a reference.
line 25: "reduction targets" - of what? carbon emissions?
Line 27: "average temperature before industrialization" mention what the value was
line 32: tense: increased
line 35: "attracted not attacked
Author Response
Point 1: "affecting factors" does not sound grammatically correct; I would refer to as "Contributing factors" or "Influential factors."
Response 1: line 2: According to the suggestion, we have corrected the "affecting factors" into "Influential factors."
Point 2: line 10: ...this paper study investigated...
Response 2: line 10: According to the suggestion, we have corrected the "paper" into "study."
Point 3: line 22: "Temperature rising" is too vague, state if you are referring to the environmental or global temperature increases.
Response 3: line 22: Thanks for your careful checks. The "Temperature rising" refers to the "Global temperature rising," We have modified it.
Point 4: Line 23: you state the increase is due to carbon emissions, presented as a fact; therefore, it requires a reference.
Response 4: Line 23: We provided references [1] to prove that " the main reason for global climate change is carbon emissions."
Point 5: line 25: "reduction targets" - of what? Carbon emissions?
Response 5: Line 25: The "reduction targets" is referring to the "greenhouse gas reduction targets." We have revised it and pointed it out in the study.
Point 6: Line 27: "average temperature before industrialization" mention what the value was
Response 6: Line 27: The World Meteorological Organization (WMO) sets the period of 1961-1990 as the base year. The global average temperature in the base year is 14 degrees (Celsius). However, the Paris Agreement did not specify the value of the global average temperature before industrialization; it emphasized the goal of "keep the temperature rise well below 2 degrees Celsius above pre-industrial levels". Therefore, according to the Paris Agreement, the study did not mention the average temperature value before industrialization.
Point 7: Line 32: tense: increased
Response 7: Line 36: Thanks for your suggestions. We feel sorry for the improper wording. We changed "increase" to "increased."
Point 8: Line 35: "attracted not attacked
Response 8: Line 39: We changed "attacked" to "attracted."

Reviewer 2 Report
Dear authors,
The manuscript entitled “Affecting factors and spatiotemporal characteristics of carbon intensity on industrial sectors in China” aims to assess the affecting factors on carbon intensity and the spatiotemporal characteristics of carbon intensity in China. The topic of the article is interesting. However, the manuscript needs some corrections to become suitable for publication. See the attached document to address the different corrections, suggestions and comments provided to improve the quality of the manuscript and make it publishable.
- Page 2, line 42. Move the text in the figure caption.
- Page 5, line 111-114. Move the text to the introduction section where you introduced the study objectives
- Page 9, line 259. Correct the figure number. It should be "Figure 2".
- Page 9, line 273. I suggest to break chart axis at 30 tonne/104 yuan to make data more visible
- Page 11, Line 307. All images must be self-explanatory. Please add a legend to indicate what are all the coloured points.
- Page 11, Line 309. All images must be self-explanatory. Please add a legend to indicate what are all the coloured points.
Author Response
Response to Reviewer 2 Comments
Point 1: Page 2, line 42. Move the text in the figure caption.
Response 1: Line 45: Thanks for your suggestions. We moved the text to the figure caption.
Point 2: Page 5, line 111-114. Move the text to the introduction section, where you introduced the study objectives.
Response 2: Line 117-212: We have explained the research content and study objectives of this study in the Introduction Section. Line 111-114 summarized the literature review and proposed our study's innovation points based on the analysis of existing literature.
Point 3: Page 9, line 259. Correct the figure number. It should be "Figure 2".
Response 3: Line 265: Thanks for your suggestions. We feel sorry for our carelessness. We changed "Figure 3" to "Figure 2".
Point 4: Page 9, line 273. I suggest to break chart axis at 30 tonne/104 yuan to make data more visible
Response 4: Line 278: Thanks for your suggestions. To make data more visible, we added a secondary axis.
Point 5: Page 11, Line 307. All images must be self-explanatory. Please add a legend to indicate what are all the colored points.
Response 5: Line 314: Thanks for your suggestions. We added a legend and indicated all the colored points.
Point 6: Page 11, Line 309. All images must be self-explanatory. Please add a legend to indicate what are all the colored points.
Response 6: Line 318: Thanks for your suggestions. We added a legend and indicated all the colored points.

Reviewer 3 Report
This paper investigated the factors that affected carbon intensity, including employee, the industry added value, fixed assets investment, coal consumption, and resource tax. The results indicated that Manufacturing of Oil Processing and Coking Processing (S7), Manufacturing of Non-metal Products (S10), Smelting and Rolling Process of Metal (S11), and Electricity, Gas, Water, Sewage Treatment, Waste and Remediation (S17) contribute most to carbon intensity in China. The topic is interesting, However, the research statements, problem, and positioning of the paper need more developments. This includes informing the reader of the present state of the knowledge in the field, why this research idea is needed, what contribution the research could make, and to which theory. However, I recommend the author explain more clearly which main theory they exactly want to adopt/explore in the conceptual model development. Also, state these theories in the Introduction, and more straightforward in Hypothesis & Methodology section. The current form of the Introduction section is short. Authors need to explain more directly their main idea. Try to focus on the research gap issue of previous studies and your motivation to tackle this problem and provide the solution. Authors should justify why they chose China for their study, the study period of 2005-2005 also needs justification.
Author Response
- Response to Reviewer 3 Comments
Point 1: the research statements, problem, and positioning of the paper need more developments. This includes informing the reader of the present state of the knowledge in the field, why this research idea is needed, what contribution the research could make, and to which theory. However, I recommend the author explain more clearly which main theory they exactly want to adopt/explore in the conceptual model development. Also, state these theories in the Introduction, and more straightforward in Hypothesis & Methodology section. The current form of the Introduction section is short. Authors need to explain more directly their main idea. Try to focus on the research gap issue of previous studies and your motivation to tackle this problem and provide the solution.
Response 1:
(1) The research statements, problem, and positioning of the study.
Line 27-31: Thanks for your suggestions. According to UNEP data, in 2018, Chinese carbon emissions ranked first globally, accounting for more than a quarter of the world's total emissions. It shows that China is facing tremendous pressure to reduce emissions. Industrial sectors are the backbone of the national economy, providing material guarantees for social production and life. At the same time, it will generate massive energy consumption, especially fossil energy. Notably, fossil energy combustion is indispensable in promoting carbon emissions.
Line 117-121:In general, the innovations of this study are in two aspects. First, we calculated carbon intensity at the level of 20 industrial sectors in China from 2005 to 2015; moreover, the extended STIPAT method be utilized to study the affecting factors of carbon intensity. Second, we employed the spatial panel econometric analysis model to examine the temporal and spatial variations of carbon intensity, additionally analyzed the direct effects and spillover effects of carbon intensity among industrial sectors.
- The present state of the knowledge in the field, why this research idea is needed, what contribution the research could make, and to which theory.
Line 55-63:The development of an industry in the economy and society will be affected by the environment. Meanwhile, the relationship between industrial sectors is also undergoing profound changes. An industry cannot exist independently of the economy and other sectors, indicating the close connections between sectors. Up to now, most of the studies have explored the spatio-temporal variations of carbon intensity from the level of national or provincial; however, few studies have paid close attention to the affecting factors and spatial characteristics of carbon intensity from an aspect of industrial sectors. Additionally, many studies have decomposed the affecting factors of the high energy-intensive industries, such as the Power industry, Transportation industry, and the Coal industry. However, the study on the spatial characteristics of carbon intensity at industrial sectors is current in the blank appearance.
- The current form of the Introduction section is short.
Line 27-31: In the introduction section, we have added some content appropriately, including the status quo of carbon emissions in China and the significance of studying carbon emissions in Chinese industrial sectors.
(4) Try to focus on the research gap issue of previous studies and your motivation to tackle this problem and provide the solution.
Line 64-71: Our study utilized the spatial panel econometric analysis model to find out the influential factors and the spatial characteristics of carbon intensity on 20 industrial sectors in China. The contributions of our study include: (1) Compared with other literature, we analyze the affecting factors on carbon intensity in the industrial sectors. (2) We expand the spatial distance matrix in the spatial analysis method to make it suitable for the spatial characteristics of industrial sectors and decomposes the affecting factors with the extended STIPAT model. Given that few studies consider the impact of environmental regulation, this paper took advantage of the resource tax to express the role of environmental regulations. (3) We attempt to explore the spillover effect of carbon intensity with the spatial panel econometric analysis model from the perspective of industrial sectors.
Point 2: Authors should justify why they chose China for their study; the study period of 2005-2005 also needs justification.
Response 2:
- Line 27-31: Chinese carbon emissions are ranking first globally, accounting for more than a quarter of the world's total emissions. It shows that China is facing tremendous pressure to reduce emissions. Industrial sectors are the backbone of the national economy, providing material guarantees for social production and life. At the same time, it will generate massive energy consumption, especially fossil energy. Notably, fossil energy combustion is indispensable in promoting carbon emissions.
(2) Line 249-255: In the actual statistical process, the wide range of product exchanges between various industries and departments requires human resources, material resources, and time. Therefore, in China, the corresponding input-output tables are only available in the years with mantissa 2 and 7, which directly results in the discontinuity of the input-output table. Due to the limitation of actual data, many studies using the input-output table to analyze practical problems can only be limited to some years. Considering that the input-output data before 2000 is too short of timeliness, this paper only uses the input-output table data after 2000. Meanwhile, the most recent year of the input-output table published by the China Input-Output Society is 2012, so the latest data used in this paper is 2015.
